# Comparison of Transforaminal Lumbar Interbody Fusion in the Ambulatory Surgery Center and Traditional Hospital Settings, Part 2: Assessment of Surgical Safety in Medicare Beneficiaries

**DOI:** 10.3390/jpm13030566

**Published:** 2023-03-22

**Authors:** Scott M. Schlesinger, Dominic Maggio, Morgan P. Lorio, Kai-Uwe Lewandrowski, Jon E. Block

**Affiliations:** 1Legacy Spine & Neurological Specialists, 8201 Cantrell Rd., Ste. 265, Little Rock, AR 72227, USA; 2Advanced Orthopedics, 499 E. Central Pkwy., Ste. 130, Altamonte Springs, FL 32701, USA; 3Center for Advanced Spine Care of Southern Arizona, 4787 E., Camp Lowell Drive, Tucson, AZ 85712, USA; 4Independent Consultant, 2210 Jackson Street, Ste. 401, San Francisco, CA 94115, USA

**Keywords:** fusion, interbody cages, stand-alone, ambulatory surgery center, Medicare, CPT ^®^ 22630, 22633, safety

## Abstract

(1) Background: The clinical benefits and procedural efficiencies of performing minimally invasive fusion procedures, such as transforaminal lumbar interbody fusion (TLIF), in the ambulatory surgery center (ASC) are becoming increasingly well established. Currently, Medicare does not provide reimbursement for its beneficiaries eligible for TLIF in the ASC due to a lack of evidence regarding procedural safety. However, the initiation of the Hospital Without Walls program allowed for traditional hospital procedures to be relocated to other facilities such as ASCs, providing a unique opportunity to evaluate the utility of TLIF in the ASC in Medicare-age patients. (2) Methods: This single-center, retrospective study compared baseline characteristics, intraoperative variables, and 30-day postoperative safety outcomes between 48 Medicare-age patients undergoing TLIF in the ASC and 48 patients having the same procedure as hospital in-patients. All patients had a one-level TLIF using the VariLift^®^-LX expandable lumbar interbody fusion device. (3) Results: There were similar patient characteristics, procedural efficiency, and occurrence of clinical 30-day safety events between the two study groups. However, there was a marked and statistically significant difference in the median length of stay favoring TLIF patients treated in the ASC (23.9 h vs. 1.6 h, *p* = 0.001). All ASC-treated patients were discharged on the day of surgery. Postoperative visits to address adverse events were rare in either group. (4) Conclusions: These findings provide evidence that minimally invasive TLIF can be performed safely and efficiently in the ASC in Medicare-age patients. With same-day discharge, fusion procedures performed in the ASC offer a similar safety and more attractive cost–benefit profile for older patients than the same surgery undertaken in the traditional hospital setting. The Centers for Medicare and Medicaid Services should strongly consider extending the appropriate reimbursement codes (CPT ^®^ 22630, 22633) for minimally invasive TLIF and PLIF to the ASC Covered Procedure List so that Medicare-age patients can realize the clinical benefits of surgeries performed in this setting.

## 1. Introduction

Significant advances in spinal surgery technology and techniques have allowed for the advent and rapid evolution of increasingly less invasive operative approaches that minimize the size of the incision and surgical “footprint” [1,2]. With the commensurate reduction in soft tissue and ligamentous disruption, minimally invasive procedures are often safer, quicker, and promote a faster recovery than traditional open-spine surgery [3].

Importantly, the recent evidence also suggests that patients treated with minimally invasive transforaminal lumbar interbody fusion (TLIF), for example, have similar two-year clinical outcomes as patients having open TLIF [4]. The less invasive characteristics of minimally invasive TLIF are also reflected in an improved health economic profile with studies demonstrating superior cost-effectiveness compared to open fusion procedures [5,6].

The technological advances achieved with minimally invasive surgery have not only given spine surgeons revolutionary ways to perform lumbar fusion [7], but they have also enabled the surgery to be performed in outpatient settings, such as Ambulatory Surgery Centers (ASC) [8,9,10,11,12,13,14,15,16,17,18]. With outcomes that are similar to those achieved in the traditional hospital setting [9,19], it has been estimated that a broader base of patients can safely be referred for lumbar fusion surgery in the ASC [18].

While advantages such as less blood loss, lower risk of infection, reduced post-operative pain and pain medication use, and faster return to daily activities are often appropriately ascribed to minimally invasive spine surgery, these approaches have also been lauded for their potential to expand the age range of patients eligible for lumbar fusion in an outpatient setting. Unfortunately, the benefits of performing these surgeries in an ASC cannot be offered to Medicare beneficiaries as the Centers for Medicare and Medicaid Services (CMS) has not extended the appropriate reimbursement codes for TLIF (CPT ^®^ 22630, 22633) to the ASC Covered Procedures List (CPL).

However, in March 2020, CMS announced the Hospital Without Walls (HWW) program, which provided broad regulatory flexibility due to the exigencies of the COVID-19 pandemic that allowed hospitals to provide services and procedures in locations beyond their existing walls, such as ASCs, while still receiving hospital payments under Medicare [20]. This scenario afforded the unique opportunity to evaluate the safety and effectiveness of minimally invasive TLIF procedures among Medicare beneficiaries treated in ASCs that enrolled as “temporary hospitals”. Herein, we report the comparative utility and safety of performing minimally invasive TLIF procedures in the ASC setting compared to the traditional hospital setting in patients 65 and older.

## 2. Materials and Methods

This single-center, retrospective study compared baseline characteristics, intraoperative variables, and near-term postoperative safety outcomes between Medicare-age patients undergoing TLIF in the ASC and those having the same procedure in the traditional hospital setting. Chronic back and/or leg symptoms and radiographic evidence of degenerative spondylosis with segmental instability in all cases necessitated surgical decompression and fusion. All 48 patients having a TLIF procedure in the ASC as a result of the HWW initiative were included in this study. As a comparison group, the last 48 Medicare-age patients undergoing TLIF in the hospital setting prior to the HWW initiative were included. All procedures were performed by one of two surgeons.

Chart review captured the patient background data including age, gender, body mass index (BMI), primary diagnosis, American Society of Anesthesiologists’ (ASA) physical status classification, and details of prior spine surgery. Patients were selected in reverse chronological order to identify all eligible cases for inclusion in this analysis. Intraoperative data included blood loss, transfusion requirements, operative time, complications, and length of stay. Thirty-day postoperative outcomes tabulated the frequency and timing of emergency department visits, hospital and intensive care unit admissions, re-operations, and adverse events including infections.

All patients had a one-level TLIF using the VariLift^®^-LX expandable lumbar interbody fusion device (Wenzel Spine, Austin, TX, USA) (Figure 1) [21]. VariLift^®^-LX is a posterior stand-alone expandable lumbar interbody fusion device cleared by the FDA (K180822) for 1 or 2 levels, PLIF or TLIF, with or without supplemental fixation, and intended for use with autograft and/or allograft tissue. All fusion procedures at both locations were conducted with general anesthesia.

The minimally invasive surgical arthrodesis technique and procedures have been detailed previously [19]. Briefly, after patient positioning, anatomical landmark identification, and incision, microscopic technique, and fluoroscopic confirmation were used, and neural decompression was performed by bilateral or unilateral laminectomy, discectomy, and preparation of the disc space for the implantation of the expandable interbody fusion device. The appropriately sized interbody device (VariLift^®^-LX) was then placed with fluoroscopic guidance, expanded, and filled with morselized autograft and/or allograft bone. All ASC cases received a stand-alone device. Only two in-hospital cases (4%) required additional fixation with pedicle screw instrumentation in conjunction with a posterolateral fusion.

All patients in both groups had the same standard postoperative care with follow-up visits at two weeks, six weeks, three months, and six months postoperatively. Postoperative radiographs were obtained in the third and sixth months.

Univariate descriptive statistics such as means, medians, and associated variability measures as well as frequency distributions were computed for all background and perioperative characteristics. All variables were compared statistically between study groups using the two-sample *t*-test (2-tailed) for continuous outcomes and Fisher’s exact test and the Mann–Whitney U-test for categorical variables as appropriate. Postoperative visits at 24 h, as well as within 7 and 30 days were tabulated as frequencies for each study group.

## 3. Results

Table 1 provides the comparative background characteristics for patients in both study groups. Aside from a greater preponderance of previous adjacent level fusions among the in-hospital patients, the groups were similar across all variables without significant differences.

Inspection of Table 2 also shows a similar magnitude and distribution of perioperative values between study groups with the exception of intraoperative blood loss and length of stay. While median blood loss was significantly greater among ASC patients, the amount of loss in either setting (200 vs. 100 ccs) was clinically negligible and similar to previous comparisons between surgeries undertaken in the ASC versus in-hospital [19].

The median length of stay for patients treated in the traditional hospital setting was approximately fifteen times greater than for ASC patients (23.9 h vs. 1.6 h, *p* = 0.001). All patients treated in the ASC were discharged well before midnight on the day of the procedure. Figure 2 illustrates comparative distributions in length of stay between study groups.

No patient in either group required a blood transfusion. There were three incidental durotomies in the ASC group and nine in the in-hospital group that were repaired directly with suturing at the time of surgery. None of these intraoperative events resulted in a postoperative complication and length of stay was unaffected. These patients required a drain left in place for five days and the drain was attached to a bile bag and not a Hemovac^®^.

No patient in either group had an emergency department visit within 24 h of discharge. Six patients in the ASC group and three in the in-hospital group were seen in the emergency department between days two and seven, respectively (*p* = 0.49). Two of the ASC patients were treated for pain control and three were seen for urinary retention and/or constipation. These were resolved with an indwelling foley catheter and/or medications. One patient presented with a fever and lower extremity swelling. A doppler study was negative for DVT. The three in-hospital patients were seen in the emergency department for fever, constipation, and weakness. The patient with a fever was examined, and an MRI was performed to rule out evidence of an abscess. The patient with constipation was resolved with medication. No adverse complications developed in any of these patients.

No patient in either group required hospital re-admission in the first 24 h after discharge. One patient in the traditional hospital group was admitted to a hospital within the first seven days and none in the ASC group. One patient in the ASC group was admitted to the hospital due to fever and chills between days 7 and 30 postoperatively and none were admitted in the traditional hospital group. An MRI of the lumbar spine was performed and showed no evidence of abscess. The patient was started on oral antibiotics due to a positive beta-strep blood test and a superficial surgical wound. No complications developed. No patient in either study group required reoperation at the surgical site or developed a postoperative infection requiring surgical care.

## 4. Discussion

For older spine patients, it remains imperative to minimize the surgical invasiveness and impact of the procedure to achieve satisfactory clinical outcomes [22]. Minimally invasive TLIF with standalone expandable cages has been gaining increasing acceptance among spine surgeons seeking to offer patients the simplest arthrodesis procedure [23,24,25]. This “less is more” approach may be particularly appealing to older patients by prioritizing less instrumentation, anatomy preservation, and spinal longevity. In patients without evidence of instability, concomitant use of posterior pedicle screw fixation can be avoided without its attendant complications [26,27].

Prior to the initiation of the CMS’ HWW program, fusion procedures such as minimally invasive TLIF performed in the ASC were limited to non-Medicare, commercially insured patients. The HWW program was established during the COVID-19 pandemic to give hospitals broad regulatory flexibility to provide services in locations beyond their facilities as a means of preserving hospital beds for the afflicted. Up until then, Medicare beneficiaries, on the other hand, were restricted to having these same procedures performed solely in the traditional hospital setting. This conundrum resulted in a classic *Catch-22* wherein CMS would not expand the list of ASC-payable services to include TLIF procedures without evidence of procedural safety in their target population, but these procedures could not be performed and the supporting data could not be garnered in older patients since Medicare would not provide reimbursement for the surgery in the ASC. The HWW initiative solved this dilemma and allowed for the current comparison of intraoperative variables and near-term safety outcomes between Medicare-age patients treated in the ASC and those treated in the hospital.

This study noted strikingly similar patient characteristics, procedural efficiency, and the rare occurrence of clinical 30-day safety events between the two study groups. We did observe a significant difference between groups in perioperative blood loss (100 and 200 ccs); however, these volumes were well within the expected range of reported values for minimally invasive TLIF of 126 ccs to 772 ccs [28,29,30,31,32]. The most notable study finding was a marked and statistically significant difference in length of stay favoring patients treated with single-level TLIF in the ASC. In a previous study, we also demonstrated a greater than 10-fold difference in length of stay between commercially insured TLIF-treated patients in the ASC versus those in the traditional hospital setting [19]. This corroboration of a similar safety profile and same-day discharge strongly underscores the necessity for establishing ASCs as an appropriate site of service for a select population of Medicare beneficiaries. Our TLIF-specific results also mirror population-based findings regarding the procedural safety of shifting surgical procedures and resources to the ASC setting. Employing a random 20% national sample of Medicare beneficiaries, Hollenbeck et al. [33] found that the opening of an ASC in a Hospital Service Area resulted in a decline in hospital-based outpatient surgery without increasing mortality or admission.

A recent report by Shahi et al. [18] highlighted the procedural and cost inefficiencies of minimally invasive TLIF procedures performed in the hospital setting. Of 71 patients eligible for treatment in the ASC, only 4% were discharged on the day of surgery when their surgeries were performed in-hospital. The median length of stay was 27 h, a value similar to the length of stay in the current study for in-hospital patients (~24 h). Most of the discharge delay was due to modifiable factors inherent in having surgical procedures in the traditional hospital setting (e.g., delayed physical therapy evaluation and clearance). They concluded that these patients would have (and should have) been managed more efficiently, without a modification to surgical technique or protocol, in the ASC.

While advancements in techniques, technologies, and efficiencies have allowed for relatively complex spinal procedures, such as TLIF, to be safely performed in the ASC, the corresponding surgical intensity has increased. Current minimally invasive approaches, utilizing small aperture access portals, requiring a precise microscopic technique, and with or without direct visualization, demand a mastery of real-time fluoroscopic or endoscopic guidance for instrumentation. Indeed, while the procedure may be more efficiently executed in the ASC, it is not safer or less difficult because of the ASC setting. The technical aspects of the procedure and the intensity of surgical work remain the same. Nowadays, the ASC environment is preferred by surgeons and their patients because of lower narcotic requirements and overall higher satisfaction with in-home recovery [34].

## 5. Conclusions

Our study findings provide supporting evidence that minimally invasive single-level TLIF can be performed safely and efficiently in the ASC in Medicare-age patients. With a substantially truncated length of stay allowing for same-day discharge, fusion procedures performed in the ASC can offer a more attractive alternative for older patients than similar surgeries undertaken in the traditional hospital setting. Our data were gathered from two surgeons at a single site which is a limitation of this study. However, our findings corroborate previous studies of the benefits and efficiencies of spine surgeries performed in the ASC [35,36]. We encourage CMS to strongly consider expanding their list of ASC-approved procedures for Medicare beneficiaries to include lumbar spine fusion procedures including minimally invasive TLIF and PLIF (CPT ^®^ 22630, 22633). In its future rulings, CMS differential payments to ASCs will not only require parity with adequate payments for surgeons’ professional fees but also account for implant carveouts to the ASC to facilitate increased TLIF utilization in the ASC setting.

## Figures and Tables

**Figure 1 jpm-13-00566-f001:**
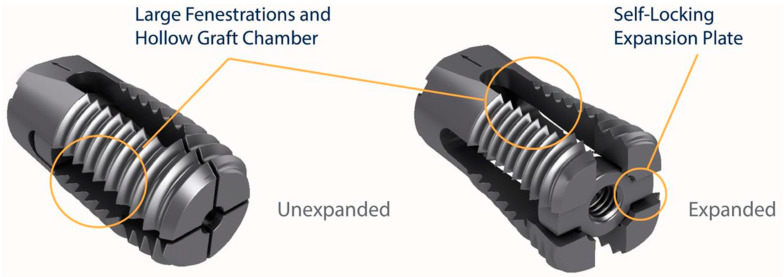
VariLift^®^-LX stand-alone lumbar interbody fusion device (Wenzel Spine, Austin, TX, USA) shown in unexpanded and expanded configurations.

**Figure 2 jpm-13-00566-f002:**
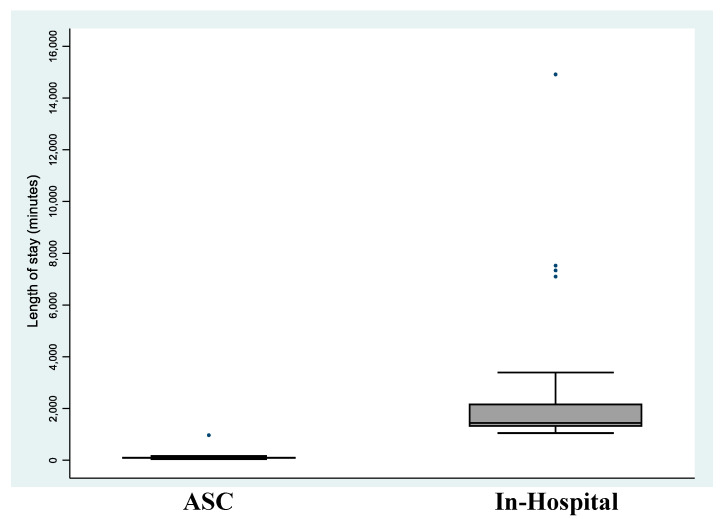
Box and whisker plot indicating length of stay by study group. The box indicates the upper and lower quartiles, and the central line is the median. The points at the end of the “whiskers” are the 2.5% and 97.5% values.

**Table 1 jpm-13-00566-t001:** Background Characteristics.

Characteristic	ASC (n = 48)	In-Hospital (n = 48)	*p*-Value
Female, n (%)	21 (44)	27 (56)	0.31
Age, mean (SD) yrs	73 (6.0)	74 (5.6)	0.43
BMI, mean (SD) kg/m^2^	28 (5.1)	29 (4.5)	0.39
ASA Grade, n (%)			
I	0 (0)	0 (0)	0.61
II	25 (53)	22 (46)	
III	22 (47)	25 (52)	
IV	0 (0)	1 (2)	
Prior Laminectomy, n (%)	15 (31)	17 (35)	0.83
Adjacent-level Fusion, n (%)	5 (10)	17 (35)	0.007

**Table 2 jpm-13-00566-t002:** Perioperative Variables.

Variable	ASC (n = 48)	In-Hospital (n = 48)	*p*-Value
Treated Level, n (%)			
L1–2	1 (2)	0 (0)	0.07
L2–3	2 (4)	4 (8)	
L3–4	3 (6)	11 (22)	
L4–5	27 (56)	21 (44)	
L5–S1	15 (31)	12 (25)	
Blood Loss, median (range) cc	200 (100–500)	100 (25–1250)	0.001
Transfusion, n (%)	0 (0)	0 (0)	1.0
Operative Duration, median (range) mins	120 (67–382)	111 (65–498)	0.29
Complications, n (%)	3 (6)	9 (19)	0.12
Length of Stay, median (range) mins	98.5 (50–969)	1436 (1053–14,914)	0.001

## Data Availability

The data presented in this study are available on request from the corresponding author. Individual participant data that underlie the results reported in this article will be made available (after deidentification) from 9 to 36 months after article publication.

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
