# Peer review of "Comparison of Transforaminal Lumbar Interbody Fusion in the Ambulatory Surgery Center and Traditional Hospital Settings, Part 2: Assessment of Surgical Safety in Medicare Beneficiaries"

_jpm, 2023, doi:10.3390/jpm13030566_

Round 1
Reviewer 1 Report
I appreciate the opportunity to review this study. The authors have conducted a study reporting the comparative utility and safety of performing minimally-invasive TLIF procedures in the ASC setting compared to the traditional hospital setting in patients 65 and older.
The manuscript is well-written and detailed; it is apparent great time and effort went into the prepared manuscript. I have some comments about the manuscript which I will list below.
- Page 2, lines 80-81: It is not clear to me who is the control (or comparison group), I suggest describing this a little more this group;
- Table 2. What kind of complications do the patients present?
- Was the satisfaction of the patients measured?
- How much experience do the surgeons have?
Author Response
Response to Reviewer 1
Page 2, lines 80-81: It is not clear to me who is the control (or comparison group), I suggest describing this a little more this group;
In the first paragraph of the Materials and Methods section we provide the following wording to describe the 2 groups:
“All 48 patients having a TLIF procedure in the ASC as a result of the HWW initiative were included in this study. As a comparison group, the last 48 Medicare-age patients undergoing TLIF in the hospital setting prior to the HWW initiative were included.”
Additionally, we have added:
“Patients were selected in reverse chronological order to identify all eligible cases for inclusion in this analysis.”
- Table 2. What kind of complications do the patients present?
As specified in the fourth paragraph of the Materials and Methods section, all intraoperative complications, in both groups, consisted of incidental durotomies that were repaired directly with suturing at the time of surgery.
- Was the satisfaction of the patients measured?
As we report on the longer-term clinical effectiveness in these patients in subsequent publications, we will report on the patient satisfaction with surgery. This manuscript focused on 30-day safety outcomes solely.
- How much experience do the surgeons have?
All surgeries were undertaken by one of two surgeons. The lead surgeon is a fellowship-trained neurosurgeon that has been managing patients with advanced spinal conditions since 1992.
Reviewer 2 Report
Thank you for inviting me to review the manuscript titled "Comparison of Transforaminal Lumbar Interbody Fusion in the Ambulatory Surgery Center and Traditional Hospital Settings, Part 2: Assessment of Surgical Safety in Medicare Beneficiaries" by Scott M. Schlesinger et al. This single-center retrospective study compares the safety of the TLIF procedure performed at ambulatory surgery centers and traditional hospitals, building on the authors' prior multicenter study published in JPM. After reviewing the manuscript, I have the following comments and concerns.
Title: The study utilized a stand-alone expandable TLIF cage technique, which enabled the insertion of the cage without additional instrumentation and reduced the surgical footprint. This technique may explain why the procedure could be performed in a day-surgery setting. Given the importance of this technique in the study, I recommend adding the keyword "stand-alone" to the manuscript's title.
Abstract: I recommend that you remove the appeal to the Medicare and Medicaid Services for the Current Procedural Terminology (CPT) codes from your conclusions section. While your study is certainly relevant to healthcare policies and regulations, the primary focus of your study was to assess the safety of the standalone TLIF procedure in ambulatory surgery centers and traditional hospital settings.
Line 106-108: What are the indications for additional pedicle screw instrumentation and posterolateral fusion? I suggest that the two cases involving these procedures be excluded from the study, and that two additional patients with a stand-alone device in the in-hospital group be added to the series. It is important to note that instrumentation and posterolateral fusion are not considered minimally invasive techniques and could significantly lengthen the in-hospital stay, potentially introducing a confounding factor that may affect the comparison between the two groups.
Table 2: The table presented in the manuscript includes a patient in the in-hospital group who lost 1250cc of blood during the procedure but did not receive a blood transfusion. This seems unlikely, and I wonder if there was a mis-coding or error in the chart review that needs to be clarified.
In addition, I suggest presenting the Length of Stay in hours instead of minutes to provide readers with a clearer concept of time.
Conclusions: I suggest including a separate paragraph that discusses the limitations of the study before the conclusion section. This will help readers to better understand the scope and implications of the study's findings.
Author Response
Response to Reviewer 2
Title: The study utilized a stand-alone expandable TLIF cage technique, which enabled the insertion of the cage without additional instrumentation and reduced the surgical footprint. This technique may explain why the procedure could be performed in a day-surgery setting. Given the importance of this technique in the study, I recommend adding the keyword "stand-alone" to the manuscript's title.
We have added “stand-alone” to the keyword list but have not altered the manuscript title as we have retained 2 patients in our data presentation that required supplemental fixation.
Abstract: I recommend that you remove the appeal to the Medicare and Medicaid Services for the Current Procedural Terminology (CPT) codes from your conclusions section. While your study is certainly relevant to healthcare policies and regulations, the primary focus of your study was to assess the safety of the standalone TLIF procedure in ambulatory surgery centers and traditional hospital settings.
The issue we address in this manuscript is extremely salient and germane to the current debate about coverage recommendations in the spine field, particularly among the Medicare population. We have caucused as an authorship group and wish to retain this language as these data will have direct implications on reimbursement for older patients treated in the ASC. We hope you will agree to retention of this language.
Line 106-108: What are the indications for additional pedicle screw instrumentation and posterolateral fusion? I suggest that the two cases involving these procedures be excluded from the study, and that two additional patients with a stand-alone device in the in-hospital group be added to the series. It is important to note that instrumentation and posterolateral fusion are not considered minimally invasive techniques and could significantly lengthen the in-hospital stay, potentially introducing a confounding factor that may affect the comparison between the two groups.
Intraoperatively it was determined that additional posterior stabilization was required to successfully manage the underlying instability observed at the operative level. We have retained these 2 patients in the analysis as we employed strict data gathering methods. We have added the following text to the second paragraph of the Materials and Methods section to clarify our methods of data capture:
“Patients were selected in reverse chronological order to identify all eligible cases for inclusion in this analysis.”
We also feel it is important for the reader to note the few times (2 of 48) the operative procedure needed to be modified intraoperatively to accommodate conditions that differed from preoperative planning.
Table 2: The table presented in the manuscript includes a patient in the in-hospital group who lost 1250cc of blood during the procedure but did not receive a blood transfusion. This seems unlikely, and I wonder if there was a mis-coding or error in the chart review that needs to be clarified.
We have re-examined the data records, and this patient did not receive a transfusion.
In addition, I suggest presenting the Length of Stay in hours instead of minutes to provide readers with a clearer concept of time.
We have provided length of stay in hours in the Abstract (23.9 hrs. vs. 1.6 hrs., p=0.001) and in the third paragraph of the Results section.
Conclusions: I suggest including a separate paragraph that discusses the limitations of the study before the conclusion section. This will help readers to better understand the scope and implications of the study's findings.
We have noted in the Conclusion section that a limitation of this study was that it was limited to the experience of two surgeons at a single clinical site.